# ON UNSUPERVISED-SUPERVISED RISK AND ONE-CLASS NEURAL NETWORKS

## ABSTRACT

Most unsupervised neural networks training methods concern generative models, deep clustering, pretraining or some form of representation learning. We rather deal in this work with unsupervised training of the final classification stage of a standard deep learning stack, with a focus on two types of methods: unsupervised-supervised risk approximations and one-class models. We derive a new analytical solution for the former and identify and analyze its similarity with the latter. We apply and validate the proposed approach on multiple experimental conditions, in particular on four challenging recent Natural Language Processing tasks as well as on an anomaly detection task, where it improves over state-of-the-art models.

## 1 INTRODUCTION

Machine learning systems often share the same architecture composed of two stages: the first stage computes representations of the input observations, while the second stage performs classification based on these representations. Most unsupervised training methods focus on the first stage: representation learning. This includes for instance generative models (VAE, GAN...), clustering techniques and, in the Natural Language Processing (NLP) domain, all recent contextual words embeddings (RoBERTa, XLNet, GPT-2...).

This work rather deals with the final classification step, more precisely how to train neural *classifiers* in an unsupervised way. In contrast to unsupervised training of the first stage that aims at learning representations, unsupervised training of the final stage may rather pursue one of the following objectives, among others:

- Training one-class models for anomaly detection
- Exploiting unsupervised approximations of the classifier risk to train a model from a priori knowledge and unlabeled data instead of labeled samples

The former is a special type of binary classification task, where the positive class represents "normal" observations and the objective is to identify unknown and often rare observations that can be considered as anomalies and form the negative class.

The latter deals with training standard discriminative classifiers without labels, i.e., when assuming that the precise target classification task is not defined explicitly with sample labels, but implicitly with a priori knowledge. We review in Section 2 the family of one-class models as well as an unsupervised approximation of the risk, and explore their relation in Section 3.3, hence bridging the gap between both unsupervised discriminative classification approaches.

The main original contributions of this work are:

- We derive an exact and analytical solution (Eq 5) to the risk approximation proposed by Balasubramanian et al. (2011)
- We analyze the properties of this solution, which lead to the following new results:
  - We extend this solution into an end-to-end differentiable loss that can be easily integrated into any modern deep learning toolkit (Eqs 6, 7)
  - We propose an unsupervised training algorithm based on this analysis (Alg 1)

– We propose a new posterior regularization term to improve this approach (Eq 8)

- We identify and study the similarity of this approximation with the one-class neural network anomaly detection method (Section 3.3)

- We validate experimentally the unsupervised model on several datasets and tasks, including a comparison with state-of-the-art one-class neural networks (Section 4)

## 2 RELATED WORK

We focus in this literature review on two unsupervised training methods for discriminative classifiers that do not aim at computing representations of the input space, but that rather exploit such representations to perform a final classification task. The first such method is an unsupervised-supervised (we have adopted the terminology of the original paper) approximation of the classifier risk that has been proposed by Balasubramanian et al. (2011) and that is detailed in Section 2.1. The second class of methods is the family of one-class models for anomaly detection, which is reviewed in Section 2.2.

### 2.1 RISK APPROXIMATION

Let be given a binary linear classifier with parameters $\theta$ that computes a scalar score $f(x) \in \mathbb{R}$ for observation $x$. The classifier outputs class $\hat{y} = 0$ iff $f(x) <= 0$, and $\hat{y} = 1$ iff $f(x) > 0$. The risk of this classifier with a hinge loss is (Balasubramanian et al., 2011):

$$R(\theta) = E_{p(x,y)} \left[ (1 - f(x) \cdot (2y - 1))_+ \right] \tag{1}$$

$$R(\theta) = P(y = 0) \int p(f(x) = \alpha | y = 0)(1 + \alpha)_+ d\alpha + P(y = 1) \int p(f(x) = \alpha | y = 1)(1 - \alpha)_+ d\alpha \tag{2}$$

Balasubramanian et al. (2011) prove that this risk can be optimized in an unsupervised way, as the labels $y$ are not required to compute Eq 1, when assuming that:

- The class-marginal prior $P(y)$ is known;
- The class-conditional distribution of the scores $p(f(x)|y)$ is Gaussian, which is supported by the central limit theorem - please refer to Balasubramanian et al. (2011) for further details.

The training algorithm proposed by the authors consists in the combination of (i) a gradient descent to optimize the linear classifier parameters $\theta$; and (ii) the Expectation-Maximization (EM) algorithm, to compute the Gaussian parameters.

We derive a new formulation of this risk and study it in Section 3.1.

### 2.2 ONE-CLASS MODELS

One-class models are based on the assumption that all observations belong to a single positive, "normal" class, except for (a few) outliers associated with the negative class. Given that there is no label to identify which observations are outliers, the problem can be cast as an unsupervised training problem. This class of models are typically used in anomaly detection applications.

The model at the origin of this research domain is the One-Class SVM (Schölkopf et al., 2001) (OC-SVM). This model projects positive observations into a feature space, and computes an hyper-plane in this feature space that separates most of these points from the region close to the origin, where outliers (noise) are assumed to be. The objective function of this model is:

$$\min_{w,r,e} \left( \frac{1}{2} ||w||^2 - r + \frac{1}{\nu N} \sum_i^N e_i \right)$$

under the constraints ($e_i$ are slack variables): $e_i \geq 0$ and $w^T \phi(x_i) \geq r - e_i$.

$w$ corresponds to the linear classifier weights and $\phi$ is the non-linear SVM projection. $w^T\phi(x) - r$ is the signed distance between any of the $N$ samples and the decision hyperplane.

A powerful extension of the one-class SVM is the Support Vector Data Description (SVDD) model (Tax & Duin, 2004). This model exploits an hypersphere with radius $R$ and center $c$ instead of an hyperplane to separate the positive and negative classes:

$$\min_{R,c,e} \left( R^2 + \frac{1}{\nu N} \sum_i^N e_i \right)$$

under the constraints that $e_i \geq 0$ and $||\phi(x_i) - c||^2 \leq R^2 + e_i$

The SVDD model has been enriched by Ruff et al. (2018) to learn a representation $\phi_W(x)$ computed with a deep neural network with parameters $W$, which gives the Deep SVDD model:

$$\min_{R,W} \left( R^2 + \frac{1}{\nu N} \sum^N \max\left(0, ||\phi_W(x_i) - c||^2 - R^2\right) + \frac{\lambda}{2} \sum_l ||W_l||^2 \right) \tag{3}$$

This model is trained by alternating a Stochastic Gradient Descent (SGD) step on $W$ and computing the optimum $R$.

Finally, the original OC-SVM has also been extended as a deep learning model with the One-Class Neural Network (OC-NN) (Chalapathy et al., 2018). In this model, the final linear layer $w$ in a stack of deep neural network layers is interpreted as defining the decision hyperplane:

$$\min_{w,V,r} \left( \frac{1}{2}||w||^2 + \frac{1}{2}||V||^2 + \frac{1}{\nu N} \sum_i^N \max\left(0, r - w^T g(V x_i)\right) - r \right) \tag{4}$$

with $V$ the previous layers that compute a representation of the input and $g()$ the previous activation. This model is trained by alternating a SGD step to update $(V, w)$ and computing the optimal $r$.

Another model of this family has recently been published: the One-class Convolutional Neural Network (Oza & Patel, 2018), but the training objective of this model departs from the previous unsupervised training objectives, as this model is trained with the standard cross-entropy loss with negative samples that are artificially generated from a Gaussian distribution centered at the origin.

## 3 UNSUPERVISED SUPERVISED RISK

### 3.1 EXACT RISK DERIVATION

Starting from Eq 1, we derive[1] a closed-form solution to compute the risk from the two Gaussian means $\mu$ and variances $\Sigma$ that model the distribution of the score $f(x)$ (we note $P(y = 0) = p_0$):

$$
\begin{aligned}
R(\mu, \Sigma) = {} & \frac{p_0}{2}(1 + \mu_0)\left(1 - \mathrm{erf}\left(\frac{-1 - \mu_0}{\sigma_0\sqrt{2}}\right)\right) + \\
& \frac{1 - p_0}{2}(1 - \mu_1)\left(1 + \mathrm{erf}\left(\frac{1 - \mu_1}{\sigma_1\sqrt{2}}\right)\right) + \\
& p_0\sigma_0^2 N(-1; \mu_0, \sigma_0) + (1 - p_0)\sigma_1^2 N(1; \mu_1, \sigma_1)
\end{aligned}
\tag{5}
$$

with

$$N(\alpha; \mu, \sigma) = \frac{1}{\sqrt{2\pi\sigma^2}} e^{-\frac{(\alpha - \mu)^2}{2\sigma^2}}$$

Balasubramanian et al. (2011) proposed to optimize the risk with finite differences. We rather propose to use the analytical solution derived in Eq 5, which has the following advantages:

- The risk value is exact and not approximated;

---

[1]See full derivation in Appendix.

- Computation of the risk is much faster using Eq 5 than with numerical approximations;
- This equation is differentiable with respect to the Gaussian parameters. We derive next another function that relates the Gaussian parameters to the model parameters $\theta$. Hence, the full risk can be directly integrated as a loss function in deep learning toolkits;
- The analytical equation can be analyzed, which leads to novel insights as shown next.

Let us plot Equation 5 as a function of $(\mu_0, \mu_1)$ in Figure 1 (left), for $p_0 = 0.1$ and $\sigma_0 = \sigma_1 = 1$.

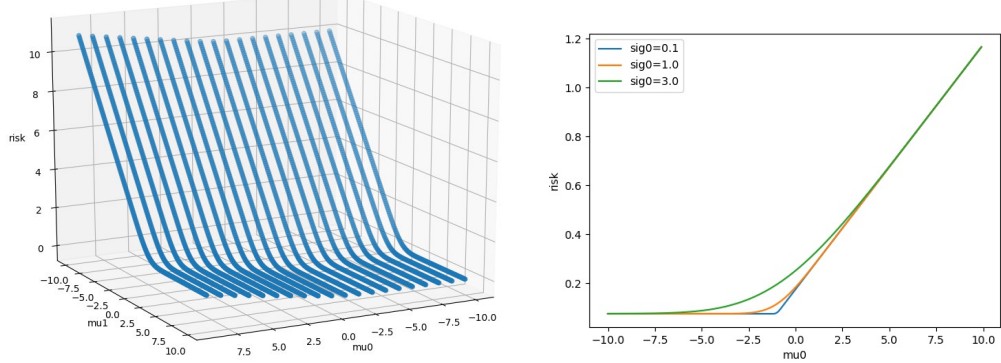

Figure 1: Risk as a function of both $(\mu_0, \mu_1)$ (left), and only $\mu_0$ (right) for $\mu_1 = 2$, $\sigma_1 = 1$ and $\sigma_0 \in \{0.1, 1, 3\}$

When we fix $\mu_1$, we can see in Figure 1 (right) that the risk as a function of $\mu_0$ can be well approximated by a scaled and translated rectified linear function, as long as the variances are small enough. Furthermore, the lower $\sigma_0$ (and $\sigma_1$) is, the better the risk is. Varying $\mu_1$ and $\sigma_1$ only translates this curve vertically, above the horizontal axis. So, assuming that the risk has first been minimized with respect to $\mu_1$, then the global minimum of the risk may be obtained by decreasing linearly $\mu_0$. Conversely, lower risks are obtained when $\mu_1$ is increasing. Although we have not exploited this piece-wise linear approximation of the risk in our implementation, it is interesting to compare it to the $\max(0, \cdots)$ term in Equation 4.

Let us now make another assumption: that both modes $(\mu_0, \sigma_0)$ and $(\mu_1, \sigma_1)$ of the score distribution are well separated. This is a reasonable assumption when we are not too far away from the global optimum, because the previous analysis has already shown that getting close to the optimum implies that $\mu_0$ is small, $\mu_1$ is large and that $\sigma_0$ and $\sigma_1$ are small. Then, a good approximation of $\mu_0$ and $\mu_1$ can be computed by splitting all the scores $f(x)$ according to the $p_0$-quantile $x_{p_0}$ defined as

$$x_{p_0} = \arg \min_x \left| p_0 - \frac{\sum_{z \in X} \mathbb{1}_{f(z) < f(x)}}{N} \right| \tag{6}$$

where the set of all observations $X$ is of size $N$. Let us call $X^-$ the subset of size $N^-$ of all data points that are on the left side of the $p_0$-quantile:

$$X^- = \{x \in X \ \ s.t. \ \ f(x) < f(x_{p_0})\}$$

and similarly for the other side:

$$X^+ = \{x \in X \ \ s.t. \ \ f(x) \geq f(x_{p_0})\}$$

We can now approximate the Gaussian parameters deterministically:

$$\mu_0 \simeq \frac{1}{N^-} \sum_{x \in X^-} f(x) \qquad \mu_1 \simeq \frac{1}{N^+} \sum_{x \in X^+} f(x) \tag{7}$$

$$\sigma_0^2 \simeq \left( \frac{1}{N^-} \sum_{x \in X^-} f(x)^2 \right) - \left( \frac{1}{N^-} \sum_{x \in X^-} f(x) \right)^2 \quad \sigma_1^2 \simeq \left( \frac{1}{N^+} \sum_{x \in X^+} f(x)^2 \right) - \left( \frac{1}{N^+} \sum_{x \in X^+} f(x) \right)^2$$

Intuitively, decreasing the risk may be achieved by decreasing $\mu_0, \sigma_0, \sigma_1$ and increasing $\mu_1$. Plugging these equations into equation 5 gives a differentiable loss with respect to the network parameters, which can be used in every modern deep learning toolkit.

## 3.2 GEOMETRIC INTERPRETATION

Following Chalapathy et al. (2018), we can consider a deep neural network that computes some representation of its inputs. These representations are then passed to a final binary linear classification layer with a single scalar output. This final layer, and optionally the previous layers, may be trained by minimizing our unsupervised risk in Eq 5 with Stochastic Gradient Descent. As discussed in Section 2.2, this final layer actually defines an hyperplane that separates both positive and negative instances, and its output is the signed distance between each observation and this hyperplane. $\mu_0$ is the average of these signed distances for all points that are on one side of the hyperplane ($X^-$), and $\mu_1$ for all points on the other side $X^+$. So decreasing $\mu_0$ and increasing $\mu_1$ can be interpreted as moving away all samples in $X^-$ and in $X^+$ as far as possible from the hyperplane, as show in Figure 2 (right).

An important constraint is that the proportion of points on both sides of the hyperplane should be equal (or close) to $p_0$, otherwise, an easy way to decrease the risk with unbalanced classes is to translate the hyper-plane along the vector $w$ infinitely, moving all samples into the most frequent class. The constraint is thus

$$f(x) \leq 0 \quad \forall x \in X^- \text{ and } f(x) \geq 0 \quad \forall x \in X^+$$

This constraint can be fulfilled by adding another term to the risk, which becomes:

$$R'(\theta) = R(\theta) + f(x_{p_0})^2 \tag{8}$$

While Balasubramanian et al. (2011) have used the class marginal only as a prior information, the additional term in Equation 8 can be seen as a posterior regularization term, which forces the posterior distribution $P(y = 0|X)$ to match $p_0$.

Algorithm 1 summarizes the training procedure.

---

**Algorithm 1** End-to-end unsupervised training

- Initialization:
  - Let consider a binary classification task, for which we assume that the proportion of class-0 elements $p_0$ is known approximately;
  - Let be given a corpus of observations $\{x_i\}_{1 \leq i \leq N}$ without labels;
  - Let be given a deep neural network $g_\phi(x)$ with parameters $\phi$ that computes a vectorial representation of an input $x$, which is fed to a final linear classification layer $f_\theta(g_\phi(x))$ with parameters $\theta$; $\phi$ and $\theta$ may be initialized randomly or pretrained.
- Iterate:
  - Run a forward pass on the dataset $\{x_i\}_{1 \leq i \leq N}$ with the current parameters $\phi, \theta$.
  - Compute all classifier scores $\{s_i = f_\theta(g_\phi(x_i))\}_{1 \leq i \leq n}$ over the full corpus $N$, or over a batch of observations $n$ that is large enough to assume that the distribution of classes in the batch is representative of the distribution in the whole corpus.
  - Sort the list of scores $(s_i)_{1 \leq i \leq n}$ to compute the $p_0$-quantile $x_{p_0}$, following Equation 6.
  - Compute the Gaussian parameters $\mu = (\mu_0, \mu_1), \Sigma = (\sigma_0, \sigma_1)$ with Equations 7.
  - Compute the risk (Eq 8) with these Gaussian parameters.
  - Apply automatic differentiation to compute $\nabla_\theta R(\mu, \Sigma)$, and optionally $\nabla_\phi R(\mu, \Sigma)$;
  - Run a step of SGD to update $\theta$, and optionally $\phi$.

---

## 3.3 RELATION WITH ONE-CLASS NEURAL NETWORKS

The One-Class Neural Network Chalapathy et al. (2018) similarly splits the set of observations with an hyper-plane defined by the last layer of a deep neural network stack, but while Equation 8 splits

the samples according to $p_0$, the OC-NN splits them according to the $\nu$-quantile of the points sorted by their signed distance to the hyper-plane, where $\nu$ controls the number of data points that are allowed to be on the negative side of the hyper-plane. This $\nu$ hyper-parameter plays the same role as our $p_0$. By rewriting their distance with our notation $f(x)$, their loss (Eq 4) becomes:

$$\min\left(L + \frac{1}{\nu}\frac{1}{N}\sum_i\left(\max(0, r - f(x_i))\right)\right)$$

where $L$ is a term that does not depend on $x_i$. Chalapathy et al. (2018) compute the optimal $r$ as "the $\nu$-quantile" of the scores, so we can rewrite $r = f(x_{p_0})$, and their sum as our $\mu_0$:

$$\min\left(L + \frac{1}{\nu}\frac{N^-}{N}\left(f(x_{p_0}) - \mu_0\right)\right) = \min\left(L + f(x_{p_0}) - \mu_0\right)$$

Their objective thus aims at maximizing $\mu_0$, i.e., making all negative samples as close as possible to the hyperplane, as shown in Figure 2 (left). This equation strongly resembles the linear approximation of the risk that we have depicted in Figure 1, except that the OC-NN takes into account only the negative part of the embeddings space, while our risk includes both negative and positive parts, and that the gradients are in opposite directions, as summarized in Figure 2.

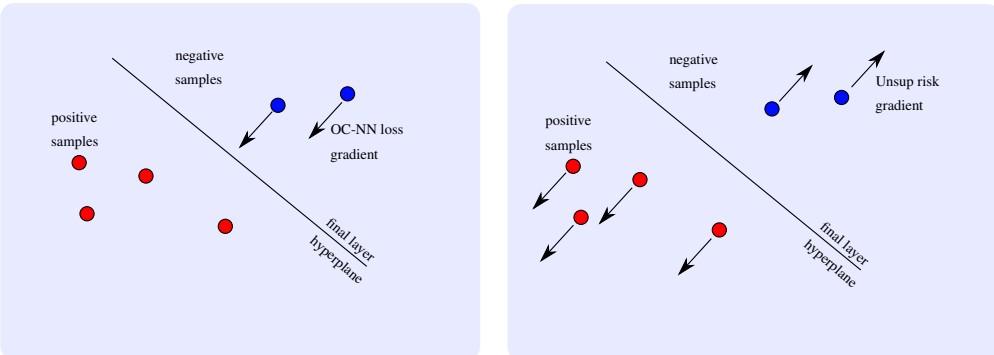

Figure 2: Comparative illustration of OC-NN and unsupervised risk approximation: observations are represented in the embeddings space just before the final linear layer; the hyperplane is defined by the parameters $\theta$, and $f_\theta(x)$ is the signed distance of the samples to the hyperplane. During training, the OC-NN tends to reduce the distance between the negative samples and the hyperplane, while our unsupervised loss tends to increase this distance for both negative and positive samples.

## 4 EXPERIMENTAL VALIDATION

The proposed unsupervised risk is coded in pytorch (Paszke et al., 2017) and is freely distributed [2] It is evaluated in various tasks: (i) on a synthetic toy classification dataset; (ii) on the Wisconsin Breast Cancer benchmark; (iii) on four NLP tasks and (iv) on a standard anomaly detection task. Following the related works, the standard unsupervised accuracy metric (Xie et al., 2016) is used for the first three cases, while the Area Under Curve (AUC) metric is used for anomaly detection.

### 4.1 SYNTHETIC DATASET

We first validate our approach on a synthetic dataset, which contains 10,000 4-dimensional instances sampled from a bi-Gaussian distribution ($p_0 = 0.6, \mu_0 = [1,1,1,1]^T, \sigma_0 = [1,1,1,1]^T; p_1 = 0.4, \mu_1 = [-2,-2,-2,-2]^T, \sigma_1 = [1,1,1,1]^T$). We train two simple models with 1,000 training epochs: one with a single layer, and another one with two layers and 2 hidden neurons (half of the input size). The accuracy per training epoch is shown on the left curve in Figure 3.

---

[2]The code is given in the supplementary material and is further distributed with an open-source license on a public git repository.

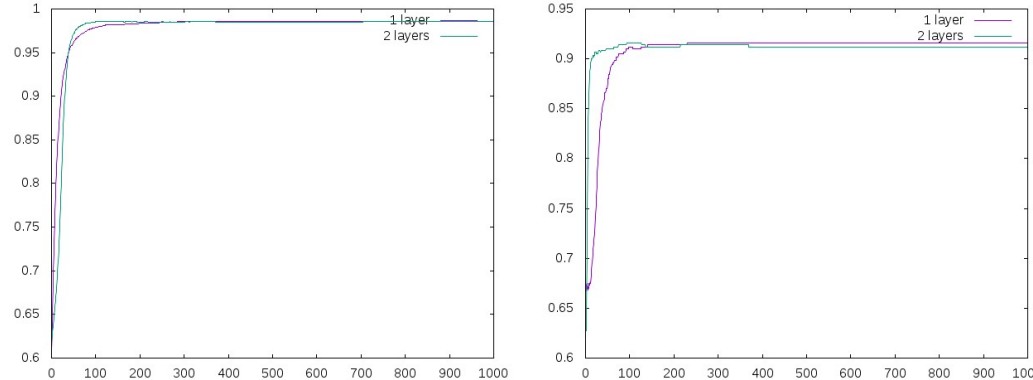

Figure 3: Accuracy as a function of the number of unsupervised training epochs on the synthetic dataset (left) and on the Wisconsin Breast Cancer dataset (right).

We further study the sensitivity of our algorithm to initial conditions by retraining the model 10 times with random initial parameters: the standard deviation of the accuracy is smaller than 2%. This first experiment validates that the unsupervised training algorithm is able to quickly and reliably converge towards the expected solution when the feature space explicitly encodes the class information.

## 4.2 WISCONSIN BREAST CANCER DATASET

We validate next our unsupervised approach on a standard machine learning benchmark for binary classification: the Wisconsin Breast Cancer dataset (Dua & Karra Taniskidou, 2017), composed of 569 instances with 30 dimensions each. The right curve in Figure 3 shows the accuracy of both our 1 and 2-layers models. 15 hidden neurons (half of the input dimension) are used for the 2-layers model.

The convergence of our method is also fast and stable on this more realistic dataset. The state-of-the-art for supervised learning on this dataset is 99.1% of accuracy (Osman, 2017). With **91%** of accuracy, our approach performs relatively well given that it is purely unsupervised. As a comparison, we have run a K-Means clustering algorithm on the same dataset, which gives **85%** of accuracy.

## 4.3 SENTEVAL TASKS

We validate next our unsupervised approach on four recent and more difficult Natural Language Processing (NLP) binary classification datasets:

- **Movie Review** (MR): classification of positive vs. negative movie reviews;
- **Product Review** (CR): classification of positive vs. negative product reviews;
- **Subjectivity status** (SUBJ): classification of subjective vs. objective movie reviews;
- **Opinion polarity** (MPQA): classification of positive vs. negative movie reviews.

These datasets as well as the experimental evaluation protocol that we have used are described in details in Conneau & Kiela (2018). This protocol first computes a sentence representation with the state-of-the-art method **InferSent** (Conneau et al., 2017), and then passes these sentence embeddings into a simple feed-forward network that is trained on each dataset.

We have adopted the same experimental protocol and the same hyper-parameters, except that we do not train the final feed-forward network with supervised labels and the cross-entropy loss, but we rather train it without any label and with our proposed unsupervised loss. Table 1 summarizes the accuracy of the state-of-the-art supervised models trained on the full corpus (*InferSent sup.*) and on only 100 instances (InferSent 100-ex), as well as the accuracy of the proposed unsupervised model (Unsup risk). Results in italic are taken from Conneau et al. (2017), other results are computed.

Table 1: Unsupervised accuracy on four binary NLP tasks

| System | CR | SUBJ | MPQA | MR |
|---|---|---|---|---|
| *InferSent sup.* | *86.3* | *92.4* | *90.2* | *81.1* |
| InferSent 100-ex | 63.8 | 62.5 | 70.1 | 53.9 |
| Unsup risk | 66.8 | **83.0** | 70.9 | 59.7 |

We can observe that the proposed purely unsupervised method always gives at least as good results as the state-of-the-art transfer learning model trained on 100 reviews, with a notable improvement of +20% absolute for the subjectivity classification task.

## 4.4 ANOMALY DETECTION

We finally validate our approach on an anomaly detection task. We adopt the same dataset and experimental protocol than Ruff et al. (2018) and Chalapathy et al. (2018) for comparison. The tasks consists in detecting outliers in digits images, where the "normal class" is the positive class and is composed of images corresponding to a single target digit, and the outliers are randomly sampled from the other digits images. Our model is composed of a single additional feed-forward layer on top of the Ruff et al. (2018) model. This final classification layer is initialized from the Ruff et al. (2018) parameters, and it is then trained in an unsupervised way with the loss in Equation 8. We tune the hyper-parameters (number of epochs and learning rate) on a development corpus obtained by keeping the same positive instances from the training corpus, but adding different negative training samples. We rerun every experiment 10 times with different seeds to compute the standard deviation. For the DeepSVDD, we report both the figures from the original paper, and the results obtained with the authors code, which may differ because of slightly varying conditions. The DeepSVDD outputs on the right are the ones that our own model is based on, and with which it should be compared to.

Table 2: Results (AUC) on anomaly detection (*: from original papers)

| | *OC-NN** | *DeepSVDD** | **DeepSVDD** | **Eq 8** |
|---|---|---|---|---|
| 0 | $97.6 \pm 1.7$ | $98.0 \pm 0.7$ | $98.0 \pm 0.6$ | $\mathbf{98.3} \pm 1.1$ |
| 1 | $99.5 \pm 0.0$ | $99.7 \pm 0.1$ | $99.4 \pm 0.2$ | $\mathbf{99.5} \pm 0.3$ |
| 2 | $87.3 \pm 2.1$ | $91.7 \pm 0.8$ | $89.2 \pm 1.8$ | $\mathbf{93.1} \pm 2.7$ |
| 3 | $86.5 \pm 3.9$ | $91.9 \pm 1.5$ | $90.5 \pm 1.5$ | $\mathbf{92.4} \pm 0.9$ |
| 4 | $93.3 \pm 2.4$ | $94.9 \pm 0.8$ | $94.0 \pm 1.3$ | $\mathbf{94.8} \pm 2.0$ |
| 5 | $86.5 \pm 3.3$ | $88.5 \pm 0.9$ | $86.3 \pm 1.3$ | $\mathbf{90.4} \pm 2.3$ |
| 6 | $97.1 \pm 1.4$ | $98.3 \pm 0.5$ | $\mathbf{98.0} \pm 0.6$ | $97.6 \pm 2.9$ |
| 7 | $93.6 \pm 2.1$ | $94.6 \pm 0.9$ | $93.7 \pm 1.4$ | $\mathbf{95.0} \pm 1.7$ |
| 8 | $88.5 \pm 4.7$ | $93.9 \pm 1.6$ | $92.7 \pm 0.9$ | $\mathbf{93.6} \pm 1.4$ |
| 9 | $93.5 \pm 3.3$ | $96.5 \pm 0.3$ | $96.0 \pm 0.7$ | $\mathbf{96.4} \pm 0.5$ |

We can note that our proposed unsupervised method always improve compared to the One-Class neural network, and is also generally better than the DeepSVDD model run on the same platform. Compared to the one-class models, our approach exploits information from all instances instead of only the negative samples (see Figure 2). Furthermore, under reasonable assumptions, our loss converges towards the theoretical optimum of the classifier risk (See Eq 1).

## 5 CONCLUSION

We have shown that both unsupervised-supervised classifier risk approximation and one-class neural networks lead to similar training procedures, although they optimize a slightly different objective. One of the main difference is that the former exploits all training samples, positive and negative, which should lead to better parameter estimates. This seems to be confirmed by experimental validation. Based on the similarity between both types of methods, we have also shown experimentally and through analysis that the unsupervised-supervised classifier risk approximation is a valuable method to be included in the set of approaches dedicated to anomaly detection. In future works, we plan to extend this approach for multi-class classification and few-shot learning.

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

## A APPENDIX

### A.1 DERIVATION OF THE UNSUPERVISED RISK

Let be given a binary linear classifier with parameters $\theta$ that computes a scalar score $f(x) = \sum_{i=1}^{n} \theta_i x_i \in \mathbb{R}$ for observation $x \in \mathbb{R}^n$. The classifier outputs class $\hat{y} = 0$ iff $f(x) \leq 0$, and $\hat{y} = 1$ iff $f(x) > 0$. The true/gold class label is noted $y \in \{0, 1\}$. The risk of this classifier with a hinge loss is (Balasubramanian et al., 2011):

$$
\begin{aligned}
R(\theta) &= E_{p(x,y)} \left[ (1 - f(x) \cdot (2y - 1))_+ \right] \\
&= P(y=0) \int p(f(x) = \alpha | y = 0)(1 + \alpha)_+ d\alpha + \\
&\quad P(y=1) \int p(f(x) = \alpha | y = 1)(1 - \alpha)_+ d\alpha
\end{aligned} \tag{9}
$$

We assume the conditional distributions follow a normal distribution:

$$p(f(x)|y = 0) \sim N(\mu_0, \sigma_0)$$

$$p(f(x)|y=1) \sim N(\mu_1, \sigma_1)$$

where $N(\mu, \sigma)$ is the standard normal distribution with mean $\mu$ and variance $\sigma^2$. We can then rewrite an approximation of this risk under the previous assumption and the additional assumption that the class priors $P(y)$ are known:

$$R = P(y=0) \int N(\alpha; \mu_0, \sigma_0)(1+\alpha)_+ d\alpha + P(y=1) \int N(\alpha; \mu_1, \sigma_1)(1-\alpha)_+ d\alpha$$

Removing the non-linearity:

$$R = P(y=0) \int_{-1}^{+\infty} N(\alpha; \mu_0, \sigma_0)(1+\alpha) d\alpha + P(y=1) \int_{-\infty}^{1} N(\alpha; \mu_1, \sigma_1)(1-\alpha) d\alpha$$

Distributing:

$$
\begin{aligned}
R &= P(y=0) \int_{-1}^{+\infty} N(\alpha; \mu_0, \sigma_0) d\alpha + P(y=0) \int_{-1}^{+\infty} \alpha N(\alpha; \mu_0, \sigma_0) d\alpha + \\
&\quad P(y=1) \int_{-\infty}^{1} N(\alpha; \mu_1, \sigma_1) d\alpha - P(y=1) \int_{-\infty}^{1} \alpha N(\alpha; \mu_1, \sigma_1) d\alpha
\end{aligned}
$$

We know that the cumulative distribution function of a $(\mu, \sigma)$ normal is:

$$F(x) = \frac{1}{2}\left(1 + \operatorname{erf}\left(\frac{x-\mu}{\sigma\sqrt{2}}\right)\right)$$

So the integral of a Gaussian is:

$$\int_a^b N(x; \mu, \sigma) dx = F(b) - F(a) = \frac{1}{2}\left(\operatorname{erf}\left(\frac{b-\mu}{\sigma\sqrt{2}}\right) - \operatorname{erf}\left(\frac{a-\mu}{\sigma\sqrt{2}}\right)\right)$$

We know that $\operatorname{erf}(-\infty) = -1$, $\operatorname{erf}(0) = 0$ and $\operatorname{erf}(+\infty) = 1$, so

$$
\begin{aligned}
R &= \frac{P(y=0)}{2}\left(1 - \operatorname{erf}\left(\frac{-1-\mu_0}{\sigma_0\sqrt{2}}\right)\right) + P(y=0) \int_{-1}^{+\infty} \alpha N(\alpha; \mu_0, \sigma_0) d\alpha + \\
&\quad \frac{P(y=1)}{2}\left(1 + \operatorname{erf}\left(\frac{1-\mu_1}{\sigma_1\sqrt{2}}\right)\right) - P(y=1) \int_{-\infty}^{1} \alpha N(\alpha; \mu_1, \sigma_1) d\alpha
\end{aligned}
$$

Integration by parts give:

$$\int_a^b x N(x; \mu, \sigma) dx = bF(b) - aF(a) - \int_a^b F(x) dx$$

We also know that

$$\int \operatorname{erf}(x) dx = x\operatorname{erf}(x) + \frac{e^{-x^2}}{\sqrt{\pi}} + C$$

So

$$\int_a^b F(x) dx = \frac{b-a}{2} + \frac{1}{2}\int_a^b \operatorname{erf}\left(\frac{x-\mu}{\sigma\sqrt{2}}\right) dx$$

We use integration by substitution:

$$\int_a^b \operatorname{erf}\left(\frac{x-\mu}{\sigma\sqrt{2}}\right) dx = \sigma\sqrt{2}\int_a^b \operatorname{erf}\left(\frac{x-\mu}{\sigma\sqrt{2}}\right) \frac{dx}{\sigma\sqrt{2}} = \sigma\sqrt{2}\int_{u(a)}^{u(b)} \operatorname{erf}(u) du$$

with $u(x) = \frac{x-\mu}{\sigma\sqrt{2}}$

So

$$\int_{u(a)}^{u(b)} \operatorname{erf}(u) du = u(b)\operatorname{erf}(u(b)) - u(a)\operatorname{erf}(u(a)) + \frac{1}{\sqrt{\pi}}(e^{-u(b)^2} - e^{-u(a)^2})$$

$$\int_{u(a)}^{u(b)} \text{erf}(u)du = \frac{b-\mu}{\sigma\sqrt{2}}\text{erf}\left(\frac{b-\mu}{\sigma\sqrt{2}}\right) - \frac{a-\mu}{\sigma\sqrt{2}}\text{erf}\left(\frac{a-\mu}{\sigma\sqrt{2}}\right) + \frac{1}{\sqrt{\pi}}(e^{-\frac{(b-\mu)^2}{2\sigma^2}} - e^{-\frac{(a-\mu)^2}{2\sigma^2}})$$

$$\int_a^b \text{erf}\left(\frac{x-\mu}{\sigma\sqrt{2}}\right)dx = (b-\mu)\text{erf}\left(\frac{b-\mu}{\sigma\sqrt{2}}\right) - (a-\mu)\text{erf}\left(\frac{a-\mu}{\sigma\sqrt{2}}\right) + \frac{\sigma\sqrt{2}}{\sqrt{\pi}}(e^{-\frac{(b-\mu)^2}{2\sigma^2}} - e^{-\frac{(a-\mu)^2}{2\sigma^2}})$$

$$\int_a^b F(x)dx = \frac{b-a}{2} + \frac{b-\mu}{2}\text{erf}\left(\frac{b-\mu}{\sigma\sqrt{2}}\right) - \frac{a-\mu}{2}\text{erf}\left(\frac{a-\mu}{\sigma\sqrt{2}}\right) + \frac{\sigma}{\sqrt{2\pi}}(e^{-\frac{(b-\mu)^2}{2\sigma^2}} - e^{-\frac{(a-\mu)^2}{2\sigma^2}})$$

Plugging into the former equation:

$$\int_a^b xN(x;\mu,\sigma)dx = bF(b) - aF(a) - \frac{b-a}{2} - \frac{b-\mu}{2}\text{erf}\left(\frac{b-\mu}{\sigma\sqrt{2}}\right) + \frac{a-\mu}{2}\text{erf}\left(\frac{a-\mu}{\sigma\sqrt{2}}\right) - \frac{\sigma}{\sqrt{2\pi}}(e^{-\frac{(b-\mu)^2}{2\sigma^2}} - e^{-\frac{(a-\mu)^2}{2\sigma^2}})$$

$$
\begin{aligned}
\int_a^b xN(x;\mu,\sigma)dx =\ & \frac{b}{2} + \frac{b}{2}\text{erf}\left(\frac{b-\mu}{\sigma\sqrt{2}}\right) - \frac{a}{2} - \frac{a}{2}\text{erf}\left(\frac{a-\mu}{\sigma\sqrt{2}}\right) + \\
& -\frac{b}{2} + \frac{a}{2} - \frac{b}{2}\text{erf}\left(\frac{b-\mu}{\sigma\sqrt{2}}\right) + \frac{\mu}{2}\text{erf}\left(\frac{b-\mu}{\sigma\sqrt{2}}\right) + \\
& \frac{a}{2}\text{erf}\left(\frac{a-\mu}{\sigma\sqrt{2}}\right) - \frac{\mu}{2}\text{erf}\left(\frac{a-\mu}{\sigma\sqrt{2}}\right) - \frac{\sigma}{\sqrt{2\pi}}(e^{-\frac{(b-\mu)^2}{2\sigma^2}} - e^{-\frac{(a-\mu)^2}{2\sigma^2}})
\end{aligned}
$$

Simplifying

$$
\begin{aligned}
\int_a^b xN(x;\mu,\sigma)dx =\ & \frac{\mu}{2}\left(\text{erf}\left(\frac{b-\mu}{\sigma\sqrt{2}}\right) - \text{erf}\left(\frac{a-\mu}{\sigma\sqrt{2}}\right)\right) - \\
& \frac{\sigma}{\sqrt{2\pi}}(e^{-\frac{(b-\mu)^2}{2\sigma^2}} - e^{-\frac{(a-\mu)^2}{2\sigma^2}})
\end{aligned}
$$

Note: another way to obtain this result is to use the following known formula:

$$\int_a^b xN(x;\mu,\sigma)dx = \mu\int_a^b N(x;\mu,\sigma)dx - \sigma^2\left[N(x;\mu,\sigma)\right]_a^b$$

So

$$\int_a^b xN(x;\mu,\sigma)dx = \frac{\mu}{2}\left(\text{erf}\left(\frac{b-\mu}{\sigma\sqrt{2}}\right) - \text{erf}\left(\frac{a-\mu}{\sigma\sqrt{2}}\right)\right) - \sigma^2\left(N(b;\mu,\sigma) - N(a;\mu,\sigma)\right)$$

When $b \to +\infty$:

$$\int_a^{+\infty} xN(x;\mu,\sigma)dx = \frac{\mu}{2}\left(1 - \text{erf}\left(\frac{a-\mu}{\sigma\sqrt{2}}\right)\right) + \sigma^2 N(a;\mu,\sigma)$$

And when $a \to -\infty$:

$$\int_{-\infty}^b xN(x;\mu,\sigma)dx = \frac{\mu}{2}\left(1 + \text{erf}\left(\frac{b-\mu}{\sigma\sqrt{2}}\right)\right) - \sigma^2 N(b;\mu,\sigma)$$

Our risk is:

$$
\begin{aligned}
R =\ & \frac{P(y=0)}{2}\left(1 - \text{erf}\left(\frac{-1-\mu_0}{\sigma_0\sqrt{2}}\right)\right) + P(y=0)\int_{-1}^{+\infty}\alpha N(\alpha;\mu_0,\sigma_0)d\alpha + \\
& \frac{P(y=1)}{2}\left(1 + \text{erf}\left(\frac{1-\mu_1}{\sigma_1\sqrt{2}}\right)\right) - P(y=1)\int_{-\infty}^1 \alpha N(\alpha;\mu_1,\sigma_1)d\alpha
\end{aligned}
$$

So

$$
\begin{aligned}
R \;=\; & \frac{P(y=0)}{2}\left(1-\operatorname{erf}\left(\frac{-1-\mu_0}{\sigma_0\sqrt{2}}\right)\right)+-\frac{P(y=0)\mu_0}{2}\left(1-\operatorname{erf}\left(\frac{-1-\mu_0}{\sigma_0\sqrt{2}}\right)\right)+ \\
& P(y=0)\sigma_0^2 N(-1;\mu_0,\sigma_0)+ \\
& \frac{P(y=1)}{2}\left(1+\operatorname{erf}\left(\frac{1-\mu_1}{\sigma_1\sqrt{2}}\right)\right)-\frac{P(y=1)\mu_1}{2}\left(1+\operatorname{erf}\left(\frac{1-\mu_1}{\sigma_1\sqrt{2}}\right)\right)+ \\
& P(y=1)\sigma_1^2 N(1;\mu_1,\sigma_1)
\end{aligned}
$$

And finally, the risk as a function of the bi-Gaussian parameters is:

$$
\begin{aligned}
R \;=\; & \frac{P(y=0)}{2}(1+\mu_0)\left(1-\operatorname{erf}\left(\frac{-1-\mu_0}{\sigma_0\sqrt{2}}\right)\right)+P(y=0)\sigma_0^2 N(-1;\mu_0,\sigma_0)+ \\
& \frac{P(y=1)}{2}(1-\mu_1)\left(1+\operatorname{erf}\left(\frac{1-\mu_1}{\sigma_1\sqrt{2}}\right)\right)+P(y=1)\sigma_1^2 N(1;\mu_1,\sigma_1)
\end{aligned}
$$

