# OpenReview forum: "On unsupervised-supervised risk and one-class neural networks"
_ICLR.cc/2020/Conference — Reject_

### Official Review · AnonReviewer1 · 2019-10-21
**Official Blind Review #1**

**Rating:** 3

**Review:**

UPDATE:
I acknowledge that I’ve read the author responses as well as other reviews.
After reading, I would keep my rating at 3 (Weak Reject), since the key reasons for my rating still hold.

####################

This work makes a connection between recently introduced one-class neural networks [8, 4] and the unsupervised approximation of the binary classifier risk under the hinge loss [1]. An explicit expression of this risk approximation is derived for the case that the prior class probabilities are known and that the class-conditional distributions of classification scores are Gaussian. This solution is then used to formulate an end-to-end differentiable loss for unsupervised binary classification which is combined with a posterior class probability regularizer to avoid trivial solutions. Finally, the paper presents an experimental evaluation on synthetic data, the Wisconsin Breast Cancer dataset, four NLP tasks, as well as on the anomaly detection task on MNIST where the proposed method slightly outperforms the two existing one-class networks [8, 4].

I think this paper makes an interesting, original connection between unsupervised-supervised risk estimation and one-class neural networks which provides a principled motivation for existing methods [8, 4] and also hints to potential flaws in their formulations, namely that OC-NN [4] and soft-boundary Deep SVDD [8] make no use of positive samples during learning (as illustrated in Figure 2). The paper is not yet ready for acceptance in my opinion, however, due to the following key reasons:
(i) The experimental evaluation is not convincing and not sufficient to assess the significance of results;
(ii) Though making this connection is interesting, the theoretical derivations presented in the paper are rather straightforward.

(i) I think the experimental evaluation is the weakest part of the paper at the moment which I find not convincing due to the lack of competitors, the use of rather simple datasets, and missing experimental details. The synthetic experiment only serves as a sanity check not giving any additional insights. As the proposed unsupervised method approximates the risk of a supervised binary classifier, I agree that it makes sense to compare to the supervised “gold standard” on binary classification tasks (Wisconsin and NLP sentiment tasks) to infer the unsupervised-supervised performance gap. However, there is no comparison to other unsupervised competitors (OC-SVM, GMM, Deep SVDD, OC-NN, etc.) to put the performance of the proposed method in these experiments into perspective (only K-Means does not establish a strong baseline). Those classification tasks further are not really a convincing use case in my mind since labels here are usually available. In contrast, I find anomaly detection to be an important application of this method, but an evaluation solely on MNIST that also lacks recent deep competitors [6] is not sufficient to assess the significance of the presented results. Moreover, from the text it seems that only the hyperparameters of the proposed method are tuned on some validation set that includes positive as well as negative samples which would be an unfair advantage and might explain the slight edge in performance. Finally, many experimental details are not reported: (ia) Are the networks used randomly initialized or pretrained? (ib) How are prior class probabilities set? (ic) What are the batch sizes (relevant for quantile estimation) (id) What score are the hyperparameters tuned on? (ie) Are negative samples in the validation set from all the anomaly classes?

(ii) The technical derivations in the paper (and the appendix) are correct but rather straightforward. The theoretical heavy-lifting is from Balasubramanian et al. [1] and this paper presents an explicit solution for the risk approximation under the assumption that the prior class probabilities are known and the class-conditional score distributions are Gaussian. I do not want to discount that making this connection and loss derivation may lead to significant results (which is left to be demonstrated experimentally), but I find the current theoretical contribution on its own not sufficient. Parts of the theoretical Section 3 could also be greatly cut in my opinion (no need to define a quantile or sample mean and variance etc.). Finally, one key property of the loss expressed in the paper is its differentiability and use with autograd, but this does not hold after adding the posterior regularization term which is based on the empirical p0-quantile, correct? (the gradient of the quantile is zero almost everywhere due to the argmin)

Apart from the two key points above, the presentation of the paper is unpolished (nested lists in the main text, etc.) and major deep anomaly detection related work [10, 6, 5, 7, 3] is missing.


####################
*Additional Feedback*

*Positive Highlights*
1. The paper makes an interesting connection between unsupervised-supervised risk approximation [1] and recently introduced one-class neural networks [8, 4] that provides a principled motivation and points to potential flaws in existing formulations.
2. I think the question how to learn neural classifiers in an unsupervised manner is original and interesting.
3. The technical derivations in the paper are rigorous and correct.

*Ideas for Improvement*
4. Make a comparison to state-of-the-art unsupervised deep anomaly detection (AD) methods.
5. Run AD experiments on more complex datasets like Fashion-MNIST, CIFAR-10, and the recently introduced MVTec [2].
6. Include major deep AD works [10, 6, 5, 7, 3] into the related work.
7. Compress Sections 1–3 (fewer lists; no need to give definitions of a quantile, sample mean and variance; etc.)
8. Motivate the non-AD experiments. Currently these appear rather constructed artificially.
9. In the NLP tasks, make a comparison to text-specific one-class classifiers [9].
10. Add a sensitivity analysis w.r.t. the prior class probability p0 to infer robustness to this parameter that seems crucial.
11. Provide guidance how to select p0 in a particular application.
12. Consistently report performance metrics with standard deviations in your tables to allow to infer statistical
significance.
13. What to do if there are no negative, but only normal samples as in fully unsupervised AD? Nevertheless make an assumption on the class prior?

*Minor comments*
14. Unordered lists should be used sparsely in a main text, stylistically speaking. Avoid nesting as in the introduction.
15. Section 2.1, first sentence (and elsewhere): “Let be given ...” is grammatically wrong. Correct would be either “Let f be a binary linear classifier ...” or “Given a binary linear classifier f ...”.
16. In Section 2.1, index the classifier $f$ with parameter $\theta$, i.e. $f_\theta$. Otherwise the risk optimization parameter $\theta$ does not even appear on the right hand side of the risk Eq. (1).
17. Combine Eqs. (1) and (2) into one equation.
18. Consistently enumerate equations throughout the paper or do not enumerate at all.
19. In Eq. (3) index $i$ misses in the sum.
20. A subsection title following a section title directly is bad style. A new major section should at least be introduced with a few sentences on what this section is about.
21. Mention that $erf$ is the Gaussian error function.
22. Center the equation in Section 3.2.
23. Plots is Figure 3 are rather poorly formatted: use thicker lines and more distinctive colors; place the legend legibly.
24. Show the average with confidence intervals over the 10 runs in Figure 3.
25. Put results, as in Section 4.2 on the Wisconsin Breast Cancer dataset rather in a table.


####################
*References*
[1] K. Balasubramanian, P. Donmez, and G. Lebanon. Unsupervised supervised learning ii: Margin-based classification without labels. Journal of Machine Learning Research, 12(Nov):3119–3145, 2011.
[2] P. Bergmann, M. Fauser, D. Sattlegger, and C. Steger. Mvtec ad–a comprehensive real-world dataset for unsupervised anomaly detection. In Proceedings of the IEEE Conference on Computer Vision and Pattern Recognition, pages 9592–9600, 2019.
[3] R. Chalapathy and S. Chawla. Deep learning for anomaly detection: A survey. arXiv preprint arXiv:1901.03407, 2019.
[4] R. Chalapathy, A. K. Menon, and S. Chawla. Anomaly detection using one-class neural networks. arXiv preprint arXiv:1802.06360, 2018.
[5] H. Choi, E. Jang, and A. A. Alemi. Waic, but why? generative ensembles for robust anomaly detection. arXiv preprint arXiv:1810.01392, 2018.
[6] I. Golan and R. El-Yaniv. Deep anomaly detection using geometric transformations. In NIPS, 2018.
[7] D. Hendrycks, M. Mazeika, and T. G. Dietterich. Deep anomaly detection with outlier exposure. In ICLR, 2019.
[8] L. Ruff, R. A. Vandermeulen, N. Görnitz, L. Deecke, S. A. Siddiqui, A. Binder, E. Müller, and M. Kloft. Deep one-class classification. In International Conference on Machine Learning, pages 4393–4402, 2018.
[9] L. Ruff, Y. Zemlyanskiy, R. Vandermeulen, T. Schnake, and M. Kloft. Self-attentive, multi-context one-class classification for unsupervised anomaly detection on text. In Proceedings of the 57th Annual Meeting of the Association for Computational Linguistics, pages 4061–4071, 2019.
[10] T. Schlegl, P. Seeböck, S. M. Waldstein, U. Schmidt-Erfurth, and G. Langs. Unsupervised anomaly detection with generative adversarial networks to guide marker discovery. In Proceedings International Conference on Information Processing in Medical Imaging, pages 146–157. Springer, 2017.

**Experience Assessment:**

I have published in this field for several years.

**Review Assessment: Checking Correctness Of Derivations And Theory:**

I carefully checked the derivations and theory.

**Review Assessment: Checking Correctness Of Experiments:**

I carefully checked the experiments.

**Review Assessment: Thoroughness In Paper Reading:**

I read the paper thoroughly.

---

> ### Author Response · Authors · 2019-11-14
> **Answer to Review #1**
>
> Thank you very much for your detailed review !
> - I agree the derivation is quite simple. But it's also the best and fastest implementation of Balasubramanian et al. paper published so far: it gives an immediate and exact value of the risk, an equation that can be studied, approximated and used in deep learning toolkits, and it enables to exhibit a connection with OC-models.
> - I agree to remove the synthetic experiments, and reduce Section 3 to make more room for detailed experimental setup and new experiments.
> - I agree anomaly detection is the most interesting application here, and I compare my work to Ruff et al. (ICML'2018), which is quite recent. Golan et al. (NIPS'2018) report better results, but with a method dedicated to images, while this work is generic.
> - The proposed approach does not assume that the training set only contains positive samples (which is difficult to guarantee in real conditions without labels), so I rather consider this as an advantage. But I agree that in that sense, it may be more related to time-series outlier-detection methods rather than image one-classification, according to (Chalapathy et al., 2019). That's why I've tested it on both types of data.
> - The gradient of the posterior regularization term is never zero, except when the median is exactly on the hyper-plane. But in general, this term represents the distance of the median to the hyperplane, and its gradient will tend to make the median closer to the hyperplane. When the median sample changes, this indeed corresponds to a discontinuity of the regularization term, but at every epoch, the local gradient exists and is in general non-null.

---

### Official Review · AnonReviewer3 · 2019-10-23
**Official Blind Review #3**

**Rating:** 3

**Review:**

The anonymous authors consider the problem of training of classifiers in an unsupervised way. They propose an extension to a one-class based approach that can do anomaly detection in an unsupervised fashion.

The main contribution is a modification of the target function for the training of one-class NN. The experiments are not convincing and the modification doesn't seem to provide much inside into representation learning and anomaly detection area.

1. Figure 3: no axis labels
2. ROC AUC is not the best quality to measure the quality of imbalanced classification problems or anomaly detection, PR AUC (average precision) is better
3. In Table 2 and other experiments, there is a comparison to only one existed method e.g. authors don't reproduce results for OC-NN
4. Table 1: why compare a supervised method to an unsupervised one and don't compare to other methods?


**Experience Assessment:**

I have published one or two papers in this area.

**Review Assessment: Checking Correctness Of Derivations And Theory:**

I assessed the sensibility of the derivations and theory.

**Review Assessment: Checking Correctness Of Experiments:**

I assessed the sensibility of the experiments.

**Review Assessment: Thoroughness In Paper Reading:**

I made a quick assessment of this paper.

---

> ### Author Response · Authors · 2019-11-14
> **Answer to Review #3**
>
> Thank you for your time and review !
> - The main contribution is providing an exact loss for the unsupervised-supervised risk, which leads to an approximated form that strongly resembles the training objective of one-class models, hence establishing a connection between these two areas of research, although they were originally developed from very different fundamental hypotheses. I think such connections are always valuable in the long term, because they give researchers different points of view to analyze and interpret the behavior of methods from these families.
> - Thank you for the PR AUC reference, however in these experiments, I had to use the same evaluation metric than the state-of-the-art papers to enable comparison with them.
> - I didn't reproduce OC-NN results because they gave nearly the same results as deep-SVDD, except for 2 digits out of 10. But I agree this is another interesting comparison to do.
> - For Table 1, I've not found any unsupervised approach in the literature to compare to on SentEval. I could of course adapt other unsupervised methods to these corpora, but this is far less convincing, as when these methods give worse results than mine, one may think it is due to not-so-finely-tuned hyper-parameters, while the supervised code and model for this specific corpora were published by their authors.

---

### Official Review · AnonReviewer2 · 2019-10-24
**Official Blind Review #2**

**Rating:** 6

**Review:**

Paper summary:

This paper proposes an algorithm to train a binary classifier without supervision, simply relying on (i) class prior, (ii) the hypothesis that class conditional classifier scores are Gaussian distributed. Experiments over sentiment classification and anomaly detection highlight the effectiveness of the approach.

Review Summary

The paper reads well and is technically correct. It proposes a simple algorithm for unsupervised training of binary classifier. Experiments are appropriate but lack baseline comparison with generative models and a thorough description of the unsupervised validation procedure. Overall, the approach is simple and it would be a good paper with the addition of baselines (mixture) and the clarification of the validation procedure.

Detailed review:

The paper is divided into two parts: a closed form solution to the problem introduced by Balasubramanian et al 2011 and an algorithm leveraging class prior for training. Both parts are clear. Since the algorithm only requires the derivative of 5 wrt model parameters, could you write this derivative (it should yield a simpler expression without erf, no?).

The experimental study could discuss the robustness wrt to the choice of p_0 and report a grid of experiments over a training set with varying true and assumed p_0. In unsupervised classification, the problem of parameter validation always occurs, e.g. for senteval, how did you select p_0 and the other parameter of the model? If you used a labeled validation set, could you report its size? Could you report the performance of a supervised system trained on a set of that size? Could you report the performance of your method when fine tuned with the label of that validation set?

For the senteval experiments, it would be interesting to report the number of labels for which the supervised and unsupervised accuracy are equal. It would be more informative than simply reporting accuracy with 100 labels. When reporting the number of labels, you need to report the size of the training and validation set combined.

As far as baseline are concerned, it would be necessary to consider generative models such as mixture models and possibly mixture model with constraints on the mixing weights (Chauveau 2013).  Adding one class SVM (RBF, polynomial kernel) and one class neural nets (Ruff, Chalapati) for SentEval would be necessary to show the advantage of your approach. A curve number of training points accuracy for all models is a must have.

You could also cite related work in adaptation to new class prior, class prior estimation, see below:

Didier Chauveau, David Hunter. ECM and MM algorithms for normal mixtures with constrained parameters. 2013. ffhal-00625285v2f

Adjusting the Outputs of a Classifier to New a Priori Probabilities: A Simple Procedure
M Saerens, P Latinne, C Decaestecker - Neural computation, 2002 - MIT Press

Semi-Supervised Learning of Class Balance under Class-Prior Change by Distribution Matching
Marthinus Christoffel du Plessis, Masashi Sugiyama, ICML12

Class proportion estimation with application to multiclass anomaly rejection
T Sanderson, C Scott - Artificial Intelligence and Statistics, 2014

Finally, I feel that the assumption that model scores are Gaussian distributed would be strongly justified if you could plot the distribution/run a statistical test on whether it is the case for a supervised model with the same architecture as yours.

**Experience Assessment:**

I have read many papers in this area.

**Review Assessment: Checking Correctness Of Derivations And Theory:**

I assessed the sensibility of the derivations and theory.

**Review Assessment: Checking Correctness Of Experiments:**

I carefully checked the experiments.

**Review Assessment: Thoroughness In Paper Reading:**

I read the paper thoroughly.

---

> ### Author Response · Authors · 2019-11-14
> **Answer to Review #2**
>
> Thank you very much for your time and recommendations.
> - I agree the algorithm is simple, but one of the main contribution is connecting two previously unrelated research domains: unsupervised risk and one-class models.
> - I will improve experimental setup description and compare with generative models.
> - You're right: the gradient of the loss is easy to derive, and I will provide it in appendices.
> - p_0 is assumed to be known in the SentEval experiments: I agree it is a strong assumption, but that is reasonable in a number of practical situations. In the MNIST experiments, p_0 is set to the same value as the "nu" parameter of the deep-SVDD model.
> - Thank you very much for the references.
> - The Gaussianity assumption is verified in nearly all of the experiments I have realized, on other corpora and on SentEval and MNIST: I have actually plotted this distribution multiple times and it was always bi-modal. I didn't put such a plot here because of space constraints, but I agree it is important to add at least in annex.

---

### Decision · Program_Chairs · 2019-12-19

**Decision:**

Reject

**Comment:**

This paper makes a connection between one-class neural networks and the unsupervised approximation of the binary classifier risk under the hinge loss. An important contribution of the paper is the algorithm to train a binary classifier without supervision by using class prior and the hypothesis that class conditional classifier scores have normal distribution. The technical contribution of the paper is novel and brings an increased understanding into one-class neural networks. The equations and the modeling present in the paper are sound and the paper is well-written.

However, in its current form, as pointed out by the reviewers, the experimental section is rather weak and can be substantially improved by adding extra experiments as suggested by reviewers #1, #2. Since its submission the paper has not yet been updated to incorporate these comments. Thus, for now, I recommend rejection of this paper, however on improvements I'm sure it can be a good contribution in other conferences.